# Habitat-Suitability Model for the Yellow Rail (*Coturnicops noveboracensis*) in the Northern Gulf Coast of Alabama and Mississippi, USA

**Kelly M. Morris** [1,2] , **Eric C. Soehren** [3] , **Mark S. Woodrey** [4,5] **and Scott A. Rush** [1,*]

1   Department of Wildlife Fisheries and Aquaculture, 775 Stone Boulevard, Mississippi State University, Mississippi State, MS 39762, USA; kelly_morris@fws.gov
2   U.S. Fish and Wildlife Service, Mississippi Ecological Services Field Office, 6578 Dogwood View Parkway, Jackson, MS 39213, USA
3   Alabama Department of Conservation and Natural Resources, State Lands Division, Wehle Land Conservation Center, 4819 Pleasant Hill Road, Midway, AL 36053, USA; eric.soehren@dcnr.alabama.gov
4   Coastal Research and Extension Center, Mississippi State University, 1815 Popps Ferry Road, Biloxi, MS 39532, USA; msw103@msstate.edu
5   Grand Bay National Estuarine Research Reserve, 6005 Bayou Heron Road, Moss Point, MS 39562, USA
*   Correspondence: scott.rush@msstate.edu; Tel.: +01-662-325-0762

**Abstract:** The yellow rail (*Coturnicops noveboracensis*) is a migratory bird of high conservation priority throughout its range and winters across the Atlantic and Gulf Coastal Plains regions of the southeastern United States. Although the winter ecology of this species has been recently explored, no studies have addressed their distribution and abundance in relation to suitable habitat capable of supporting this species during winter along the northern Gulf Coast of Alabama and Mississippi. The objectives of this study were to develop a habitat-suitability model for yellow rail wintering in the northern Gulf Coast of Alabama and Mississippi. We then used this model to evaluate the distribution of habitat suitable for supporting yellow rail in this geographic area. Using a multivariate approach that makes use of presence-only data through a maximum entropy framework we compared the distribution of where the focal species was observed to a reference set of the whole study area. Of the 784,657 ha over which our model was applied, only 1% (8643 ha) of this area was predicted suitable in its present condition, for supporting yellow rail in winter. Our analysis indicates that the yellow rail along the northern Gulf Coast of Alabama and Mississippi occupy a very narrow range of environmental conditions highlighting need for specific management actions to maintain and conserve suitable winter landscapes for this habitat-restricted species.

**Keywords:** core area; *Coturnicops noveboracensis*; habitat-suitability; home range; maximum entropy; pine savanna

---

## 1. Introduction

The yellow rail (*Coturnicops noveboracensis*) is the second smallest of the North American rails [1]. This bird is highly secretive leaving much of its ecological requirements poorly understood [1–4]. Because of a small population size, limited wintering range, and threats to the disparate ecosystems this bird uses at various periods during the annual cycle, the yellow rail is listed as a Species of Conservation Concern in Canada [5], and a Migratory Nongame Bird of Conservation Concern in the United States [6].

During the non-breeding season, yellow rail can be found from the Atlantic and Gulf Coastal Plain physiographic regions of the United States from North Carolina to Texas [1]. Studies conducted on

yellow rail on their wintering grounds are limited to coastal Texas [7,8], Mississippi and Alabama [3,4], coastal South Carolina [9], and Oklahoma [10,11], with most providing anecdotal information on the occurrence of this species, not on their distribution within these states. Although historical accounts describe yellow rail overwintering along the Gulf Coast of Alabama and Mississippi, no systematic efforts have been made to survey for this species across this region with explicit aim of identifying the distribution of ecological communities that can support this species in their current condition.

Although the most unifying feature of habitat used by yellow rail in winter is dense herbaceous cover, less than 1 m in height, with little to scattered woody vegetation they are known to make use of a variety of habitats in winter, including, but not limited to, coastal high marsh, coastal prairies, rice fields, open grasslands, and wet pine savanna [1,3]. For instance, Wayne's observations in South Carolina [12] describe yellow rail winter habitat as "a low wet place of open land with a dense growth of short dead grass." In Oklahoma, Butler et al. [10] and Heck and Arbour [13] described yellow rail using habitat dominated by grasses, mainly *Sporobolus* spp., fall panic grass (*Panicum dichotomoflorum*) and tall dropseed (*Sporobolus compositus*, formerly *S. asper*).

Historically, pine savanna extended along much of the northern Gulf Coast, including coastal Alabama and Mississippi [14]. These pine savanna systems afforded conditions that supported an array of organisms, some of which are now considered threatened or endangered at state and federal levels [3,4]. Coastal pine savannas are characterized by clumps of sparsely distributed trees with open ground cover dominated by warm-season grasses [14,15]. Pine savannas were historically maintained by natural disturbance such as warm-, or growing- season fires, seasonal flooding, and herbivory [16,17]. Longleaf pine (*Pinus palustris*) ecosystems once dominated the coastal plains of the southeast United States covering approximately 24−36 million hectares, an area that extended from Virginia to Texas [18]. Today, however, longleaf pine systems are among the most ecologically degraded of all forested ecosystems [19,20], with an area currently estimated to be less than 1.2 million hectares [21]. Several factors have contributed to the decline of longleaf pine forests including agriculture and farming, habitat fragmentation, deforestation, and habitat degradation due to a lack of periodic disturbance [19,22,23]. Research suggests that the most significant contribution to the decline of pine savanna is the lack of natural disturbance, most notably fire [21].

Fire maintains pine savanna habitats by enhancing species diversity of herbaceous ground cover, suppressing the encroachment of hardwood trees and woody shrubs [24–26]. Fire may be one of the most important factors explaining the presence of yellow rail on their breeding and wintering grounds [2–4]. Given a limited winter distribution, continued loss of pine savanna and the dearth of information on how yellow rail use this habitat during the non-breeding season, our objective was to develop a habitat-suitability model (HSM) for yellow rail within a portion of the northern Gulf of Mexico, from western Mississippi to western Alabama.

For species such as the yellow rail, understanding of their distribution and habitat requirements are important elements of conservation biology [27,28]. Habitat-suitability modeling techniques that require presence-only data are increasingly used to model organismal distributions [29,30]. One such method known as maximum entropy (MaxEnt; [31]) compares the distribution of where the focal species has been observed to a reference set of the whole study area. This analysis, utilizing a multivariate statistical approach combined with the technology of geographic information systems (GIS), provides researchers and managers the tools to underpin the link between known species presence and available ecological communities, particularly the ability to map suitable habitat for focal species. By developing habitat-suitability models, our intent is to provide a map of the spatial distribution of remaining ecological communities potentially suitable for yellow rail within coastal Alabama and Mississippi, as well as how these factors may interact to support other community members throughout the northern Gulf of Mexico.

## 2. Materials and Methods

Following methods defined by Morris et al. [3] and Soehren et al. [4], we conducted surveys for yellow rail in four conservation areas (defined as study sites) in Jackson County, Mississippi: (1) Mississippi Sandhill Crane NWR (7810 ha); and (2) Jackson County Mitigation Bank (MB; 193 ha); and in Mobile County, Alabama: (3) Grand Bay Savanna Forever Wild Complex (GBS; 2164 ha) South Bog; and (4) Laurendine Tract (803 ha) owned by the Mobile County Commission (Figure 1). We surveyed for yellow rail in 19 survey sites representing 18 wet pine savanna patches ranging in size (1.9–83.3 ha). Survey sites are defined as separate management units located within the four study sites.

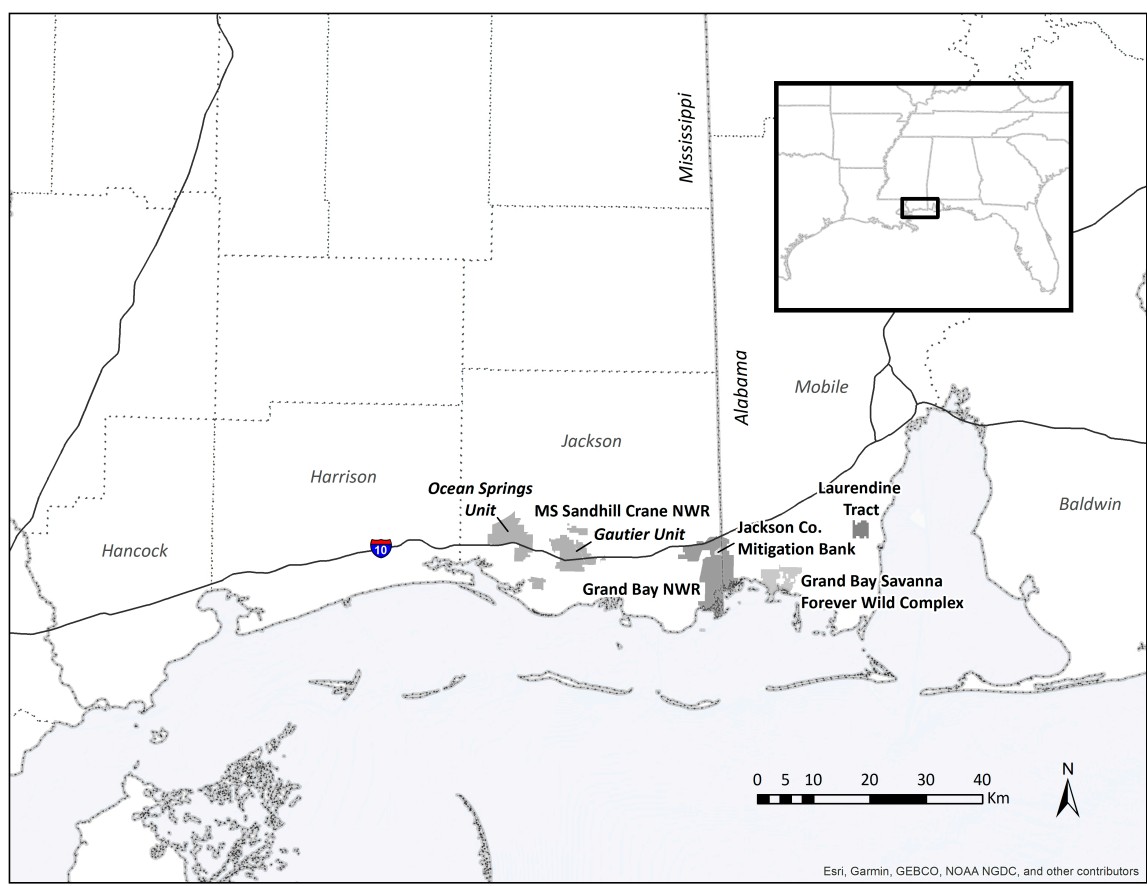

**Figure 1.** Study area for modeling the distribution of yellow rail (*Coturnicops noveboracensis*) and study sites where presence data for yellow rail was collected in Alabama (Mobile County) and Mississippi (from east to west: Jackson, Harrison, and Hancock Counties), USA.

We modeled the potential distribution of yellow rail along the East Gulf Coastal Plain in Mississippi (Harrison, Hancock, and Jackson Counties) and Alabama (Mobile County; Figure 1). These areas are characterized by low topography and infertile, acidic clay soils that often remain saturated [32], and have a temperate climate with hot summers and mild wet winters [33]. The study area contains some of the largest remaining fragments of wet pine savanna including: Grand Bay Savanna Forever Wild Complex, Alabama; the Grand Bay National Wildlife Refuge (NWR)/National Estuarine Research Reserve, and the Mississippi Sandhill Crane NWR, Mississippi [34].

Primary habitat types within the study sites are wet pine savanna, pine flatwoods, pine scrub shrub and estuarine marsh habitats. Primary understory grasses include wiregrass (*Aristida stricta*), muhly grass (*Muhlenbergia* spp.), bluestem grasses (*Andropogon* spp.), panic grasses (*Panicum* spp.), and toothache grass (*Ctenium aromaticum*). Primary herbs and forbs include *Balduina* spp., *Coreopsis*

spp., *Helianthus* spp., *Solidago* spp. and a variety of carnivorous plants (e.g., *Sarracenia*, *Drosera*, *Utricularia* spp.), with orchids and lilies scattered among the dominant graminoids [14]. Dominant woody vegetation consists of gallberry (*Ilex glabra*), wax myrtle (*Morella cerifera*), sweetbay magnolia (*Magnolia virginiana*), St. John's-wort (*Hypericum brachyphyllum*), yaupon (*Ilex vomitoria*) and various greenbriers (*Smilax* spp.). Species composition across pine savanna habitats is primarily influenced by differing soil moisture gradients [14,35,36].

Observations of yellow rail were collected using nocturnal dragline surveys as outlined by Morris [37], Morris et al. [3], and Soehren et al. [4]. We also used radio-telemetry to collect additional presence data from 20 individuals among seven study plots within our study sites (following methods described by Morris [37]). We filtered all presence locations to eliminate spatial autocorrelation among occurrence points [38]. Presence data was spatially filtered at 100 m with the Spatial Rarefy Occurrence Data tool in the SDMtoolbox for ArcGIS [39]. To account for sampling bias, we spatially filtered occurrence data to reduce the potential of repeated individuals in the model. [37]. We selected a spatial filtering distance of 100 m to represent the home range of yellow rail (3 ha) in pine savanna in Mississippi and Alabama [3]. This tool removes presence locations within a specified distance from one another which otherwise may bias models and inflate accuracy measurements [38].

MaxEnt compares conditions at species occurrence locations to randomly selected locations within a background area [40]. We captured environmental conditions throughout the study area using 10,000 randomly generated background sites, the default setting in the MaxEnt software program [41]. We selected 33 environmental variables to model the potential distribution of yellow rail in Alabama and Mississippi. Raster maps for all environmental variables were generated with the same resolution (30 m × 30 m), extent, and geographic coordinate system (GCS_North_American_1983). We used raster-based inventories of land cover derived from the 2010 Coastal Change Analysis Program (C-CAP) [42]. The C-CAP database includes 24 land cover and land use classes: developed high intensity, developed medium intensity, developed low intensity, developed open space, cultivated crops, pasture/hay, grassland, deciduous forest, evergreen forest, mixed forest, scrub/shrub, palustrine forested wetland, palustrine scrub/shrub wetland, palustrine emergent wetland, estuarine forested wetland, estuarine scrub/shrub wetland, estuarine emergent wetland, unconsolidated shore, bare land, water, palustrine aquatic bed, estuarine aquatic bed, tundra and snow/ice. We combined deciduous forest, evergreen forest, and mixed forest layers into a single class, "forest." We then selected 10 landscape variables from the C-CAP dataset we considered significant to the ecology of yellow rail, these include: grassland, forest, scrub/shrub, palustrine forested wetland, palustrine scrub/shrub wetland, palustrine emergent wetland, estuarine forested wetland, estuarine scrub/shrub wetland, estuarine emergent wetland, and water (Table 1).

For each of the 10 land cover classes we derived two landscape variables: Euclidean distance to the nearest grid cell (m) and relative frequency (0–1.0), producing a total of 20 landscape variables (Table 1). Distance layers were generated using the Euclidean distance tool in ArcGIS Pro. Frequency raster maps were generated to represent a radius of three 30 m × 30 m grid cells using the Biomapper module CircAn [29]. We selected a radius of three grid cells to represent the average home range size of yellow rail in Alabama and Mississippi (100 m, 3 ha; [30]).

We theorized that wetland soils are important in predicting the occurrence of yellow rail as these soils are more prone to hold water. Soil moisture, one of the main factors that determines the growth of grasses and trees within savanna systems is most influenced by soil characteristics and rainfall [43]. We developed one raster map representing potential wetland soil landscapes (PWSL). We expressed each grid cell in our map as a percentage of the map unit that meets the PWSL criteria. These data were generated using the Gridded Soil Survey Geographic (gSSURGO) by State [44]. To assess the significance of precipitation on the distribution of yellow rail throughout the study area, we developed 12 raster maps representing average monthly rainfall. These data were generated using PRISM Climate Data: 1981–2010 Monthly Average Precipitation by State [45]. All parameters used in our models are listed and defined in Table 1.

**Table 1.** Environmental variables used in developing species distribution models for yellow rail (*Coturnicops noveboracensis*) in Alabama and Mississippi, USA. Variables derived from same dataset are shown together with source dataset listed for first entry of data type.

| | |
|---|---|
| 1. Distance to estuarine emergent wetland | Euclidean distance to Coastal Change Analysis Program (C-CAP) land cover type. Derived from the Coastal Change Analysis Program 2010 Regional Land Cover dataset [42]. |
| 2. Distance to estuarine forest wetland | |
| 3. Distance to estuarine shrub/scrub wetland | |
| 4. Distance to forest | |
| 5. Distance to grassland | |
| 6. Distance to palustrine emergent wetland | |
| 7. Distance to palustrine forest wetland | |
| 8. Distance to palustrine shrub/scrub wetland | |
| 9. Distance to shrub/scrub | |
| 10. Distance to water | |
| 11. Frequency of estuarine emergent wetland | Frequency of C-CAP land cover type 100 m radius. Cover derived from the Coastal Change Analysis Program 2010 Regional Land Cover dataset [42]. |
| 12. Frequency of estuarine forest wetland | |
| 13. Frequency of estuarine shrub/scrub wetland | |
| 14. Frequency of forest | |
| 15. Frequency of grassland | |
| 16. Frequency of palustrine emergent wetland | |
| 17. Frequency of palustrine forest wetland | |
| 18. Frequency of palustrine shrub/scrub wetland | |
| 19. Frequency of shrub/scrub | |
| 20. Frequency of water | |
| 21. Wetland soil landscapes (PWSL) | Gridded Soil Survey Geographic (gSSURGO) by State [44]. |
| 22. Precipitation in January | Monthly precipitation averaged from 1981–2010. Derived from PRISM Climate Data: 1981–2010 Monthly Average Precipitation by State [45]. |
| 23. Precipitation in February | |
| 24. Precipitation in March | |
| 25. Precipitation in April | |
| 26. Precipitation in May | |
| 27. Precipitation in June | |
| 28. Precipitation in July | |
| 29. Precipitation in August | |
| 30. Precipitation in September | |
| 31. Precipitation in October | |
| 32. Precipitation in November | |
| 33. Precipitation in December | |

Rather than specifying a set of models a priori, that are then ranked by fit (see [46,47]), we followed the guidance of Warren et al. [30], in that we specified a range of complexity then allowed MaxEnt to control parameterization of the models. Variable inclusion was tested through a reiterative process using stepwise removal of least contributing variables [30]. To limit overfitting and improve predictive performance in our models we evaluated two sources of model complexity, multicollinearity and regularization multiplier (RM; also referred to as the β multiplier), using the R package MaxentVariableSelection [48]. Regularization multiplier determines how closely the projected distribution fit the training data, where a smaller RM results in a more localized prediction, and a large RM results in a more generalized distribution. We developed a set of models by selecting a variable contribution threshold of ≥5%, a correlation coefficient of ≤ 0.7 (Pearson's *r*), and we tested RMs from 1.0 to 6.0, in increments of 0.5. Models were run with hinge features only. Hinge features allow simpler and more succinct approximations of the true species response to the environment [40]. We selected the model with the lowest sample size corrected Akaike Information Criteria [46,47] to identify the optimal RM and environmental variables used to project the potential distribution of yellow rail throughout the study area.

We estimated the potential distribution of yellow rail using correlative ecological niche models compiled with the program MaxEnt v3.4.1 [40,41]. Potential distribution was averaged over 20 replicate models using the selected environmental predictors, RM and feature class from the top-ranking model from the variable selection analyses. Models were run using the complementary log-log (cloglog) transformation to produce an estimate of occurrence probability [31]. For each run, we selected 75% of the presence locations to develop the model, and the remaining 25% was used as training data. We assessed the final model (averaged across 20 replicates) by threshold-independent (i.e, area under the receiver operating characteristic curve, AUC) and threshold-dependent omission rate. We evaluated various AUC values (AUC training, AUC test, and AUC difference) to ensure adequate model performance. Model accuracy is considered excellent if AUC is between 0.9 and 1, good if between 0.8 and 0.9, fair if between 0.7 and 0.8, poor if between 0.6 and 0.7, and failed if AUC is between 0.5 and 0.6 [49]. We reported relative importance of environmental variables using percent variable contributions and response curves.

We projected the potential distribution of yellow rail in ArcGIS Pro by classifying occurrence probability in four categories, ranging from 0–1, derived from our final model: 'unsuitable' is $0 \leq 0.2$; 'low potential' is $0.2 \leq 0.4$; 'moderate potential' is $0.4 \leq 0.6$; and 'high potential' is $0.6 \leq 1$ [50–52].

## 3. Results

We collected 532 yellow rail presence locations from the following sources: (1) observations collected using dragline surveys between December 2012 and March 2018 at Mississippi Sandhill Crane NWR (*n* [number of unique yellow rail observed] = 74), Jackson County MB (*n* = 6), GBS South Bog (*n* = 15), and the Laurendine Tract (*n* = 2); and (2) radio telemetry locations collected between December 2012 and March 2013 at Mississippi Sandhill Crane NWR (18 individuals; *n* = 375) and Jackson County MB (2 individuals; *n* = 60; Figure 1).

Filtering using the Spatial Rarefy Occurrence Data tool, we thinned presence locations from *n* = 532 to *n* = 67. Six models had an RM = 1. The best fitting model using the selection algorithm in MaxEnt had a hinge feature, 6 uncorrelated variables (17 parameters) (Tables 2 and 3). Variables included in this model were frequency of emergent palustrine emergent wetland, frequency of shrub/scrub wetland, distance to palustrine emergent wetland, mean precipitation in June, mean precipitation in August, and mean precipitation in September. Relationships of these variables in predicting the occurrence of yellow rail at locations are shown in Figure 2. This model had an average AUC test value of 0.99 (SE = 0.006 (Table 3)). The AUC difference between training and test AUC was low (0.005), indicating that the model was not affected by overfitting toward yellow rail presence locations. Threshold-dependent measures indicated that the model had low overfitting and high discriminatory ability at the 10% omission rate (0.093; value that excludes the 10% of locations with the lowest predicted values) and lowest presence threshold (0.025; minimum predicted value for any grid cell).

**Table 2.** Species distribution model settings and evaluation metrics for yellow rail (*Coturnicops noveboracensis*) in Alabama and Mississippi, USA. SE indicates standard error of mean.

| Features [a] | β [b] | Test AUC [c] | Training AUC [d] | AUC Difference [e] | Mean Omission Rate [f] | Mean Minimum Omission Rate [g] | Number of Parameters [h] |
|---|---|---|---|---|---|---|---|
| H | 1 | 0.99 ± 0.006 SE | 0.991 ± 0.003 SE | 0.005 | 0.093 | 0.008 | 17 |

[a] H = Hinge. [b] β = Regularization multiplier. [c] Mean Area under the receiver operating characteristic plot (AUC) of testing data. [d] Mean AUC of testing data. [e] Difference between mean training AUC (calculated on training locations) and mean testing AUC (tested on evaluation locations). [f] The 10% omission rate of the training locations (value that excludes the 10% of locations with the lowest predicted values). [g] Lowest presence threshold (minimum predicted value for any grid cell). [h] Number of parameters included in the model.

**Table 3.** Percent contribution of environmental variables for the species distribution model of yellow rail (*Coturnicops noveboracensis*) in Alabama and Mississippi, USA.

| Environmental Variable | Percent Contribution | Variable Description and Source |
|---|---|---|
| Frequency of palustrine emergent wetland | 33.1 | Frequency of palustrine emergent wetland habitat within a specified radius. Derived from the Coastal Change Analysis Program 2010 Regional Land Cover dataset [42]. |
| Frequency of palustrine shrub/scrub wetland | 21.8 | Frequency of palustrine shrub/scrub wetland habitat within a specified radius. Derived from the Coastal Change Analysis Program 2010 Regional Land Cover dataset [42]. |
| June precipitation | 16.9 | Monthly precipitation averaged from 1981–2010. Derived from PRISM Climate Data: 1981–2010 Monthly Average Precipitation by State [45]. |
| August precipitation | 10.6 | Monthly precipitation averaged from 1981–2010. Derived from PRISM Climate Data: 1981–2010 Monthly Average Precipitation by State [45]. |
| September precipitation | 10.5 | Monthly precipitation averaged from 1981–2010. Derived from PRISM Climate Data: 1981–2010 Monthly Average Precipitation by State [45]. |
| Distance to palustrine emergent wetland | 7.1 | Euclidean distance to palustrine emergent wetland habitat. Derived from the Coastal Change Analysis Program 2010 Regional Land Cover dataset [42]. |

The predicted distribution map for yellow rail indicated that areas with high occurrence probability accounted for 2436 ha (0.31% of the study area); moderate probability accounted for 1356 ha (0.17%); low probability accounted for 4851 ha (0.62%); and 776,014 ha (98.9%) was considered unsuitable (Figure 3 and Table 4).

**Table 4.** Area estimates of species distribution classes predicting potential occurrence of yellow rail (*Coturnicops noveboracensis*) in Alabama (Mobile County) and Mississippi (Jackson, Harrison, and Hancock Counties), USA.

| Species Distribution Classes | Estimated Area (ha) | Proportion of the Study Area (%) |
|---|---|---|
| High potential ($0.6 \leq 1.0$) | 2436 | 0.31 |
| Moderate potential ($0.4 \leq 0.6$) | 1356 | 0.17 |
| Low potential ($0.2 \leq 0.4$) | 4851 | 0.62 |
| Unsuitable ($0 \leq 0.2$) | 776,014 | 98.9 |

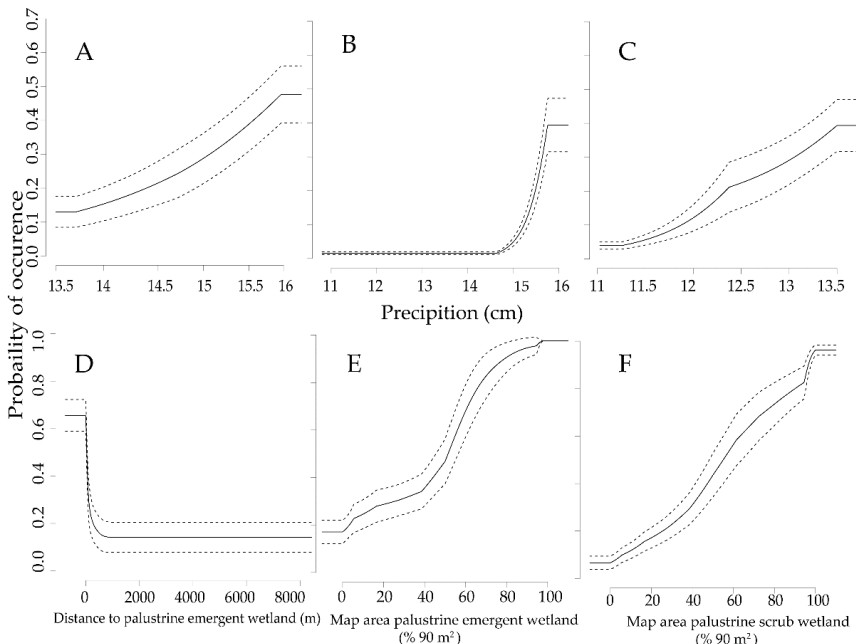

**Figure 2.** Response curves for environmental variables modeled in MaxEnt for the predicted distribution of yellow rail (*Coturnicops noveboracensis*) in Alabama and Mississippi, USA. (**A**) shows response to precipitation in June; (**B**) response to precipitation in August; (**C**) response to precipitation in September; (**D**) response to distance to palustrine emergent wetland; (**E**) response to unit of mapped area of palustrine emergent wetland (per map cell of 90 m²); (**F**) response to unit of mapped area of palustrine scrub wetland (per map cell of 90 m²). Dashed lines reflect two standard deviations of estimates.

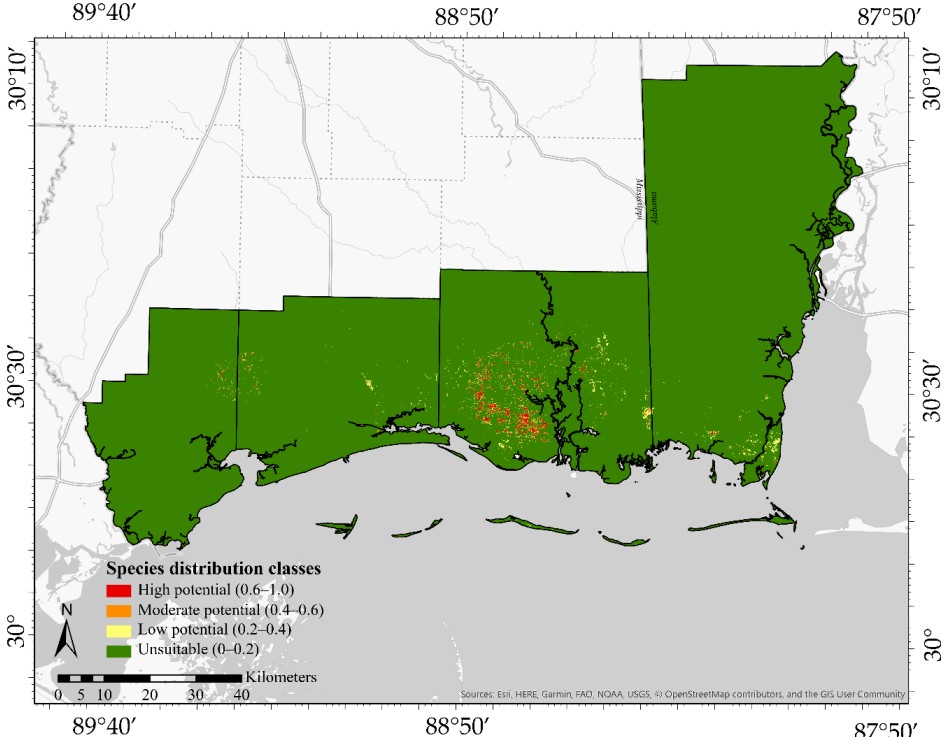

**Figure 3.** Predicted potential distribution of conditions suitable for supporting yellow rail (*Coturnicops noveboracensis*) in Alabama (Mobile County) and Mississippi (from east to west: Jackson, Harrison, and Hancock Counties), USA.

## 4. Discussion

Our results indicate the majority of the area of historical pine savanna habitat throughout our study area in Alabama and Mississippi is unsuitable for yellow rail. Historically, pine savannas were the dominant habitat type throughout our study area along the Gulf Coast of Alabama and Mississippi [14]. However, we found that 99% of our study area is considered unsuitable for yellow rail in its present condition, reflecting major changes in land cover and other environmental conditions within this geographic area.

The yellow rail occupies sites under frequent burn regimes (<3-year intervals) although other environmental factors such as seasonal precipitation and surface hydrology may also be important influences. The presence of water within pine savanna can be linked to precipitation in June through September, as highlighted in our models. They are also closely tied to emergent wetland habitats. However, hydrology alone is not enough to maintain pine savanna in conditions sufficient to support yellow rail [53]. Frequent fire regimes as well as hydrology are necessary to suspend successional progression to maintain a diverse graminoid/forb dominance characteristic of wet pine savannas [23,35,54,55]. These conditions are needed to support yellow rail [3,4], as well as other grassland-obligate species endemic to these ecosystems [18,23,36].

It is anticipated that climactic conditions will change within our study area over the relatively short term [56,57]. Such changes include increased variability in precipitation events with increased precipitation, and longer periods between precipitation events [58]. Disturbance to surface soils can also impose changes in hydrology that can affect soil saturation and residence time of surface waters [59]. Our best-supported model illustrates a link between precipitation in late summer and fall and the likelihood pine savanna systems will support yellow rail. As climate uncertainty increases and surface hydrology is altered through human development of coastal systems, these areas may experience wetter conditions and increased potential for longer periods without precipitation [58]. These meteorological extremes will affect when and where fire can be employed in managing coastal pine savanna and the hydrophytic relationships that influence the distribution of yellow rail and other pine savanna associates. Applying predictive and explanatory models such as those employed here will be key to future conservation for these species and their communities (ecosystem).

## 5. Conclusions

Our models indicate palustrine wetland habitat parameters derived from the Coastal Change Analysis Program Regional Land Cover dataset [42] are important environmental features predicting the potential occurrence of the yellow rail in Alabama and Mississippi. However, one of the caveats in developing our models was our inability to model habitat management (e.g., fire management) in predicting potential conditions suitable for supporting the yellow rail. Without proper management to maintain the ephemeral structure favorable for yellow rail presence, the regional availability of suitable habitat will ultimately depend on the application of onsite management practices that fall within the discretion and objectives of local landowners.

Large-scale conditional changes can affect the ability to maintain ecosystems in historic contexts with respect to localized management [16,60,61]. Although our suitability predictions are specific to current conditions, and while conservation mandates persist on public lands such as the Mississippi Sandhill Crane NWR and the Grand Bay Savanna Forever Wild Complex, etc., our predictive HSM provides new opportunities to strategically identify and approach other stakeholders to expand conservation stewardship within the northern Gulf-coast region. Such efforts would also benefit the entire suite of floral and faunal species associated with coastal wet pine savanna ecosystems.

Frequent fire return intervals (<3 years) in wet pine savannas are an important disturbance regime required to maintain a diverse graminoid-dominance structure in wet pine savannas and is a significant factor predicting the occupancy of a site by yellow rails [3]. Without a regular fire management regime, the open canopy grass dominant understory of pine savannas will shift toward a more-woody dominant community. As a result, these systems become less attractive to yellow

rail [3]. Wet pine savannas within coastal Alabama and Mississippi are representative of the palustrine emergent wetland and palustrine scrub/shrub wetland land cover/land use classifications of the C-CAP, which were both identified as the most important variables for predicting the distribution of the yellow rail. However, these conditions should be viewed as ephemeral without the regular application of fire.

The lack of suitable habitat within the four coastal counties of Alabama and Mississippi indicate the need for further research on the winter ecology of the yellow rail. While this study provides an illustration of the use of pine savanna systems by yellow rail, it also emphasizes the limited extent of suitable habitat within this region of their winter range.

**Author Contributions:** Conceptualization, K.M.M., E.C.S., and M.S.W., and S.A.R.; methodology, K.M.M., E.C.S., and M.S.W., and S.A.R.; formal analysis, K.M.M., E.C.S., and S.A.R.; data curation, K.M.M., M.S.W., E.C.S., and S.A.R.; writing—original draft preparation, K.M.M., E.C.S., and M.S.W., and S.A.R.; writing—review and editing, K.M.M., E.C.S., and M.S.W., and S.A.R.; visualization, K.M.M., and E.C.S.; project administration, K.M.M., E.C.S., and M.S.W., and S.A.R.; funding acquisition, E.C.S., and M.S.W., and S.A.R. All authors have read and agreed to the published version of the manuscript.

**Funding:** This research was supported by the National Institute of Food and Agriculture, U.S. Department of Agriculture, McIntire-Stennis project, grant number MISZ-082100.

**Acknowledgments:** We would like to extend our appreciation to K. Baker, P. Barnhart, M. Boone, F. Carney, P. Coppola, A. Dedrickson, K. Diamond, D. Dortch, J. Feura, L. Gardella, B. Garmon, A. Haffenden, E. Harrity, S. Hereford, J. Hintermister, H. Horne, C. Jones, T. Knudson, A. Leggett, A. Lehmicke, Z. Loman, D. McKee, L. Billodeaux-Mobray, T. Mullins, J. Murphy, S. Nesbit, W. Oakley, A. Patterson, A. Peters, T. Petroelje, E. Pulis, E. Spadgenske, B. Summerour, C. Threadgill, J. Timmer, J. Trent, J. Walker and C. Wilton for assisting in field surveys and other aspects making this study possible. Thanks to J. Trent for producing the Figure 1 map, H. Horne for providing technical assistance, and T. Sanchez for approving field survey efforts and subsequent use of data collected on the Laurendine Tract.

**Conflicts of Interest:** The authors declare no conflict of interest.

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
