# Peer review of "Habitat-Suitability Model for the Yellow Rail (Coturnicops noveboracensis) in the Northern Gulf Coast of Alabama and Mississippi, USA"

_remotesensing, doi:10.3390/rs12050848_

Round 1

Reviewer 1 Report

Please see attached file for detailed comments and suggestions for authors.

Remote Sensing- 697999- Peer Review:

This study assesses the suitability of habitat for yellow rail across coastal Alabama and Mississippi. The authors develop a model to evaluate the distribution of suitable habitat for yellow rail, and then map viable habitat across the region. While this could be a simple yet valuable contribution to the science, there are several issues with this analysis that the authors need to address or clarify before I feel that it is publishable. I have listed a summary of my major concerns below, as well as additional line by line comments.

Summary of Major Comments:

1.The authors need to clarify their sampling approach in a number of areas. I have some concerns with the number of sample sites they have relative to their analysis, which I address in the line by line comments below. Briefly, from what I can gather from their methods section, they only sample 1 site in high marsh areas, while they conduct much more extensive sampling in wet pine savanna. I’m not sold that they can make inferences about high marsh areas across such a broad area after looking at a single sampling site of this ecosystem type. Why not focus your study on the suitability of wet pine savanna areas, where the majority of your sampling took place?

2.I have some strong concerns about the authors creation of candidate models predicting yellow rail presence they input into AICc. Perhaps it is simply miscommunication, but as written, it appears the authors employed data dredging to develop and/or select their habitat models, which is well known to lead to spurious results. The authors should provide a list of their candidate models and provide ecological explanation as to why they have selected their variables and created their models. One should never just test ‘all model combinations’ (Line 197).

3. I don’t see any reporting of the AICc results or a citation for supplemental material on where to find them. This should all be reported. These are highly relevant results that have been brushed over in the methods and excluded in the results.

4. The authors discussion lacks substance as currently written. While I’m sure they have some interesting results, the discussion is mostly a presentation of facts from the scientific literature. I feel this section needs to be re-written to interpret the study’s results, show how the study’s results align with current literature, and show how the study results can push our current understanding of theory/management forward.

5. There is a lack of consistency in the objectives of the paper from the introduction to the discussion. I get very lost when the authors start the discussion with ecological niche marginality and specialization. If identifying these factors was an objective of the study, it needs to be stated in the introduction, explained in the methods, and iterated in the results to be understandable in the discussion. I think it could be a neat idea, but there needs to be consistency for it to be understandable. Otherwise, this needs to be removed from the manuscript.

Line by Line Comments:

Lines 19- 22: This portion of the abstract is misleading. The authors first state that while yellow rail winter ecology has been recently explored, their abundance and distribution relative to suitable habitat has not been explored throughout their winter range. They then go on to say that they assess the northern Gulf Coast. This doesn’t compose the entire yellow rail winter range according to their description in the introduction, which states their winter range spans from Texas to North Carolina. This could mislead a reader to believe the results represent habitat suitability for the entirety of yellow rail winter range. Please rectify.

Lines 44-48: Clarification is needed. The authors state that studies conducted on the rail have been limited to coastal Texas, coastal South Carolina, Oklahoma, Mississippi, and Alabama. But then say that they have not been conducted in the Gulf Coast of Alabama and Mississippi. Perhaps specify where in Mississippi and Alabama they have been conducted, the same way you specify that assessments have been done in coastal Texas? Also why is it important to do a survey along the Gulf Coast of Alabama and Mississippi, if surveys have already been conducted in both states? Please elaborate.

Lines 55-57: I suggest making this sentence the topic sentence of the paragraph. Currently, the paragraph seems misplaced in the introduction. The paragraph needs to align with the author’s objectives.

Lines 58-75: Its unclear why these are two paragraphs. The topic sentence of the second paragraph doesn’t align with the content or the objectives of the paper. I suggest the authors re-work these paragraphs to focus on making meaningful topic sentences. Currently, it is easy for a reader to get lost in facts and loose sight of the objectives of the paper.

Line 86: What does “with appreciation to the conservation of these ecological systems” mean?

Lines 105-112: It is unclear whether the authors assessed 19 survey sites at each study site (conservation area) or whether these survey sites were dispersed among each of these conservation areas.

Further, it needs to be clarified why they surveyed such an extensive number of wet pine savanna sites in comparison to high marsh sites. Was it simply the availability of the high marsh sites?

I’m unconvinced that a single high marsh site will be sufficient to deduce yellow rail use of high marsh habitat characteristics that could then be extrapolated to deduce the suitability of all high marsh sites along the coast. Please clarify. Should your models only really be representative of yellow rail use of wet pine savanna? Also, why do you draw a distinction between these ecosystem types here, but not in your models?

Lines 123-126: Are the values (n=x) representative of the number of surveys or the number of animals recorded?

Line 124: Please add a citation or explain drag line survey. Also is it drag line or dragline (line 129)?

Line 130: Delete ‘techniques’

Line 133-135: What was removed from the net by hand?

Lines 130- 135: It is difficult to understand whether the dragline surveys were used to collect location data, used to attach radio transmitters, or both. Please clarify your approach for data collection. Perhaps breaking the description of telemetry data collection and surveys into two separate paragraphs would be useful?

Line 150: Was presence data spatially filtered based on each individual? Or across all data? Were repeated measures on individuals accounted for in your models?

Lines 176-181: Provide ecological explanation as to why measuring these variables in each of these ways may be ecologically relevant.

Lines 182-184: Why might wetland soil landscapes be important to yellow rail and thus ecologically relevant to include in your models?

Lines 185-187: Why might precipitation be important to yellow rail? Why would each individual month be ecologically important? For instance, why might yellow rail respond specifically to precipitation in February?

Line 215: I’m not sure if it is standard practice, but because it is only two words, I would just write out regularization multiplier rather than use an acronym. There are a lot of acronyms in the paper already.

Line 193: I am not familiar with all the methods that the authors are employing, but I am confused as to why you are developing candidate models based on variable contribution thresholds? Can you add additional explanation as to how your candidate model sets were derived? You should have ecological relevance behind each of your variables and candidate models before you input them into AICc. Data dredging to look for the best variables/models from all possible combinations is incorrect and a dangerous practice. It has been well published for decades that data dredging can lead to spurious results. The authors should provide a table with their candidate model set and provide some explanation as to why they chose their variables/models.

Line 197: ‘All model combinations’ makes it sound as though the authors employed data dredging. This is well known to lead to spurious results. Please provide explanation of how you developed your candidate models using ecological knowledge.

Line 238-240: Repeating these results that are already presented in the table is redundant. I suggest removing these from the text or removing your table.

Lines 267-275: This discussion of marginality and specialization comes out of nowhere. This was not mentioned in the introduction, it is unclear why this relates to the results as written, and there is no clear indication of how this aligns with the authors objectives. I think all of this should be removed or the paper needs to be re-written to highlight that identifying these characteristics was an objective of the paper.

Lines 272: What are marginality coefficients? This wasn’t in the methods or results.

Line 274: What is a tolerance value? This wasn’t in the methods or results.

Line 276-282: This paragraph needs to be tied to your results. I appreciate that you are acknowledging there are limitations to your models, which there are. But how would this have affected your results. What does this mean for potential error? Could you be over or underestimating the extent of yellow rail distribution?

Line 283- 292: It is unclear how any of this information is related to the results of this manuscript. As currently written, this is just a literature review. There needs to be a connection in the discussion between your results and current literature. How are your results building on what is currently known or how do they align with what is currently known?

 I’m particularly interested in the authors focus on the importance of high marsh in the discussion, when it seems they had such limited sampling in this ecosystem type. I am unsure that they can say much about high marsh habitats from their analysis. Please clarify.

Line 297: As mentioned above, there are no results presented showing what your best supported models were from AICc.

Line 306-308: But this analysis doesn’t assess management application? How is this the main conclusion from your assessment?

Figures:

Figure 1: Perhaps it is the version of the map I got in my manuscript draft, but the labels on the map and the latitudes and longitudes are illegible. Even the scale bar is difficult to see. Please increase the size of the figure and the labels.

Figure 2: The colouring in these figures makes them very difficult to see. I suggest light grey confidence bands (if that’s what the blue is, it isn’t reported, please add) and a black line. The dark blue makes it impossible to see the red line. Also, please increase the axis label size. They are illegible.

In addition, in plot A, it looks as though the interesting part of the relationship is occurring within the first 500m, but it is very difficult to make out. Could you reduce the x-axis limit so this relationship could be more easily viewed?

Figure 3: Again, I suggest removing the dark blue. It washes out all of the other colours. While I appreciate the high, moderate, and low colouring matching MTBS severity classification colouring, perhaps change ‘moderate’ potential to orange and make ‘unsuitable’ light yellow? Or perhaps even just add some transparency to the blue colouring so that it lightens it? Because ‘unsuitable’ is the dominant colour, it would be easier if it was washed out a bit. Also, as in the first map, labels need to be increased everywhere. They are very difficult to read.

Reviewer 2 Report

Abstract

It is confusing to refer to the study as “northern Gulf Coast” when the model is only for AL and MS. Consider changing wording to something like “coastal MS and AL.”

Line 27 – Consider including the percent of area that was considered highly suitable.

Introduction

Line 35 – remove comma

Line 36-37 – this sentence is clunky; consider revising

Line 37 – Change “due to” to “because of”

Line 42 – Add comma after “season”. Remove “along a geographic extent”

Line 49 – Either “The yellow rail makes use” or “Yellow rails make use”

Line 50 – replace comma with m-dash after “mark”

Line 54 – again, either “yellow rails” or “the yellow rail” – consider revising this grammatical issue throughout the manuscript. I don’t believe that “rail” is a plural form.

Line 56-57 – mixing single and plural forms

Line 58 – should be “clumps of or sparsely distributed trees”

Line 58 – It may not be clear to many readers why you are suddenly talking about pine savannas here. Consider using a helpful transition phrase in the intro sentence to this paragraph or at the end of the previous paragraph.

Line 76 – Is understanding distribution and habitat not important for species that aren’t secretive? Consider rewording.

Line 86 – Can you be more specific than “within our study area”? Not clear yet the extent of this.

Materials and Methods

Line 90 – Perhaps consider moving this paragraph to later in the Methods. First step is to define the study area, 2nd is to find birds, third step is develop and define a model extent, and fourth step is to describe the model.

Line 123-128 – This paragraph should be in the Results section.

Line 129 – this paragraph is a bit clunky and could be streamlined.

Line 150 – This paragraph is somewhat repetitive and could be streamlined.

Line 151 – Should this be 90 m2? 90 x 90 = 8100, which is approximately 0.75 ha. I assume this is the reasoning for the scale chosen, but is it?

Line 155-156 – This is a Result.

Line 157-160 – Only 67 occupied points versus 10,000 randomly generated sites? Any indication that this creates problems with model fit. If so, how was this addressed?

Line 161 – I’m not clear how you got to 33 variables based on the subsequent two paragraphs.

Line 181 – Here you use a home range size of 3.37 ha, but earlier (line 151), you use the core area of 0.75 ha. Why the inconsistency?

Line 185-187 – One map for each month? If so, summer months (when Yellow Rails aren’t present) are used because of their influence on warm-season vegetation productivity, which could carry-over into winter habitat quality?

Results

The Results lack a summary of how many sites were occupied, by how many birds, etc.

Table 1 – why not include additional models in this table?

Lines 238-242 – This is somewhat repetitive with the first paragraph. Consider condensing and streamlining.

Figure 2 – Describe the blue shading. 95% CI?

Figs 2c and 2d – Should the x-axis be in a metric unit?

Discussion

Line 267 – “northern Gulf of Mexico” is sometimes defined as TX to FL. Perhaps consider revising here (and throughout) to coastal MS and AL.

Line 272 – again, the plural of rail is “rails”, so change to “yellow rails” or “the yellow rail” throughout.

Line 273 – add comma after “habitats”

Line 277 – So how does fire influence the C-CAP habitat definitions?

Line 283 – Consider revising the intro sentence to include a statement about high marsh and savanna. This paragraph could be more about how these habitat types are interconnected, and what that means for Yellow Rails.

Line 310 and 317 – change “while” to “although”

Line 316-317 – I don’t think the lack of suitable habitat indicates the need for more research. The lack of suitable habitat indicates the need for conservation and management, which may in turn need additional research to do correctly.

Round 2

Reviewer 1 Report

The flow of the introduction and discussion has greatly improved. I appreciate the clarification in the methods and the inclusion of additional citations. As someone who is unfamiliar with some of the author’s methods, I am able to much more easily follow what was done. The figures are also greatly improved.

I only have a few comments for the authors.

Comments:

Line 49: “his species” to “this species”

Line 205-210: I feel like this section would better fit under the second paragraph in the methods. I don’t want to push stylistic preference, but I do feel like it still requires an additional sentence of clarification. The authors state “n[number of unique yellow rail observed]” on line 207, but then say on line 209 that with radio telemetry they had 18 individuals (which is technically n based on their definition) and then “n=375” which is, I assume, the number of recorded locations? In their analysis, I assume they are treating these locations as independent individuals after filtering the data. I know this is already eluded to in the methods, but it would be nice to have additional clarification in this paragraph.

Line 266: Are your percentages of suitable habitat based only on historical pine savanna? Since you previously were also looking at high marsh sites, which I assume could also be suitable habitat that you did not assess, are those excluded from your calculations of unsuitable habitat? This should be clarified in your methods and results.

Author Response

February 21, 2020

Dear Reviewer,

Please consider the attached second revision of our manuscript Habitat-suitability model for the Yellow Rail (Coturnicops noveboracensis) in the northern Gulf Coast of Alabama and Mississippi, USA  (manuscript ID 697999) for publication in Remote Sensing. I greatly appreciate your comments on an earlier draft of this manuscript and the opportunity to revise it.

My co-authors and I all appreciate the speed of your review as well as the comments provided. These comments helped clarify and greatly improve the overall value and contribution of this manuscript. In revising the manuscript please find that we have implemented each of the changes suggested. We provide specific responses to your comments below (your comments paraphrased where applicable in regular font, followed by our responses in italics).

We greatly appreciate the chance to resubmit to Remote Sensing. We hope you will find this version of the manuscript to be much improved.

Sincerely,

Scott Rush

Cc: Kelly Morris, Eric Soehren, and Mark Woodrey

We’ve addressed each of the reviewer’s major comments below. As per the reviewer’s report we’ve separated these comments among the various sections of the manuscript, starting with the Abstract.

Line 49: “his species” to “this species”

Author’s Response: We’ve made this change. This revision can now be found on line 49 of the manuscript.

Line 205-210: I feel like this section would better fit under the second paragraph in the methods. I don’t want to push stylistic preference, but I do feel like it still requires an additional sentence of clarification. The authors state “n[number of unique yellow rail observed]” on line 207, but then say on line 209 that with radio telemetry they had 18 individuals (which is technically n based on their definition) and then “n=375” which is, I assume, the number of recorded locations? In their analysis, I assume they are treating these locations as independent individuals after filtering the data. I know this is already eluded to in the methods, but it would be nice to have additional clarification in this paragraph.

Author’s Response: As suggested we’ve moved this paragraph to a new location in the methods, providing additional text for clarify. We’ve also reworded some of this paragraph so that it is more straight-forward and less confusing. This paragraph now appears in lines 124 – 134 of the manuscript.

Line 266: Are your percentages of suitable habitat based only on historical pine savanna? Since you previously were also looking at high marsh sites, which I assume could also be suitable habitat that you did not assess, are those excluded from your calculations of unsuitable habitat? This should be clarified in your methods and results.

Author’s Response: We’ve reworked the text in the Methods section to afford additional clarity on the percentages of the total area that are now identified as comprised as the various levels of suitability outlined in our paper. As we no longer considered high marsh in our analysis, this ecological community type was not included in our assessment.
